# Association of Heart Rate Variability with Pulmonary Function Impairment and Symptomatology Post-COVID-19 Hospitalization

**DOI:** 10.3390/s23052473

**Published:** 2023-02-23

**Authors:** Estelle A. M. C. Adang, Maud T. A. Strous, Joop P. van den Bergh, Debbie Gach, Vivian E. M. van Kampen, Roel E. P. van Zeeland, Dennis G. Barten, Frits H. M. van Osch

**Affiliations:** 1Department of Emergency Medicine, VieCuri Medical Centre, 5912 BL Venlo, The Netherlands; 2Department of Internal Medicine, VieCuri Medical Centre, 5912 BL Venlo, The Netherlands; 3NUTRIM School of Nutrition and Translational Research in Metabolism, Maastricht University, 6229 ER Maastricht, The Netherlands; 4Department of Clinical Epidemiology, VieCuri Medical Centre, 5912 BL Venlo, The Netherlands; 5Department of Pulmonology, VieCuri Medical Centre, 5912 BL Venlo, The Netherlands; 6Department of Epidemiology, Maastricht University, 6229 ER Maastricht, The Netherlands

**Keywords:** heart rate variability, COVID-19, post-COVID-19 condition, long COVID, autonomic dysfunction, vagal nerve activity, symptomatology, lung function

## Abstract

The persistence of symptoms beyond three months after COVID-19 infection, often referred to as post-COVID-19 condition (PCC), is commonly experienced. It is hypothesized that PCC results from autonomic dysfunction with decreased vagal nerve activity, which can be indexed by low heart rate variability (HRV). The aim of this study was to assess the association of HRV upon admission with pulmonary function impairment and the number of reported symptoms beyond three months after initial hospitalization for COVID-19 between February and December 2020. Follow-up took place three to five months after discharge and included pulmonary function tests and the assessment of persistent symptoms. HRV analysis was performed on one 10 s electrocardiogram obtained upon admission. Analyses were performed using multivariable and multinomial logistic regression models. Among 171 patients who received follow-up, and with an electrocardiogram at admission, decreased diffusion capacity of the lung for carbon monoxide (DLCO) (41%) was most frequently found. After a median of 119 days (IQR 101–141), 81% of the participants reported at least one symptom. HRV was not associated with pulmonary function impairment or persistent symptoms three to five months after hospitalization for COVID-19.

## 1. Introduction

As of November 2022, over 639 million cases of SARS-CoV-2 have been reported to the World Health Organization (WHO), leading to 6.6 million deaths [1]. Since mortality rates and the severity of disease have seemingly declined because of increasing population immunity and the availability of an adequate vaccination program, the focus is shifting towards the long-term consequences of COVID-19 infection. Although a worldwide uniform case definition is lacking, long-term sequelae of COVID-19 are often referred to as post-COVID-19 condition (PCC) or long COVID. The WHO has defined PCC as ‘persistence of symptoms for at least two months occurring within three months after COVID-19 infection which cannot be explained by an alternative diagnosis’ [2].

The most frequently reported symptoms include fatigue, post-exertional malaise, shortness of breath and cognitive dysfunction [2,3]. A meta-analysis including 1655 hospitalized patients showed that 76% reported at least one symptom six months after discharge [4]. The persistence of symptoms for a prolonged period is associated with reduced quality of life [5,6,7] and work capacity [3,8]. These findings indicate that PCC not only has a devastating effect on the individual’s well-being, but also comes with a negative social and economic impact. Early recognition and adequate intervention may help to lower the burden of symptoms and to reduce their impact on quality of life. The risk of developing PCC rises with increasing age and is associated with female gender [9,10,11]. However, it is unknown which patients are particularly at risk for developing PCC, partly because the pathophysiology is not fully understood.

An acute COVID-19 infection has been shown to activate the sympathetic nervous system, thereby inducing pro-inflammatory cytokine release and subsequent cytokine response storm [12]. The vagus nerve is an important neuro-immuno-modulator and suppresses inflammation by stimulating the hypothalamic–pituitary–adrenal axis, which in turn leads to the release of the inflammation-suppressing hormone cortisol [13]. Additionally, vagal efferent fibers stimulate the spleen to secrete acetylcholine, which subsequently suppresses the release of pro-inflammatory cytokines by splenic macrophages [14,15,16]. In case of decreased vagal activity, the anti-inflammatory regulatory response is diminished, leading to excessive inflammation causing severe complications from COVID-19 [17,18]. Recent studies suggest that persistent autonomic dysfunction may also be responsible for the long-term consequences of COVID-19 [12,19]. Possible pathophysiological mechanisms explaining dysautonomia in patients with PCC include neurotropism, procoagulative state and inflammation [20]. However, it remains unclear whether dysautonomia is caused by the autonomic virus pathway or immune-mediated processes after viral exposure [20].

A few studies with a small sample sizes have shown that patients suffering from PCC had a range of functional and/or structural alterations in the vagus nerve, supporting the existence of a correlation between PCC and autonomic imbalance in which vagal nerve activity is decreased [20,21]. The activity of the vagus nerve can be indexed by the measurement of heart rate variability (HRV).

Therefore, the aim of this study was to assess whether HRV measured upon admission is associated with persistent symptoms and pulmonary function impairment between three and five months post-COVID-19 hospitalization.

## 2. Materials and Methods

### 2.1. Study Population

This was a single-center retrospective cohort study in a teaching hospital in The Netherlands. All consecutive patients admitted to regular hospital wards because of COVID-19 (defined as positive PCR assay) between February and December 2020 were eligible for inclusion. Patients admitted to the intensive care unit (ICU) were excluded because of potential overlap with PCC [22]. This time frame represents the first and second COVID-19 wave in The Netherlands. Patients were included if they were older than 18 years and completed a standardized outpatient follow-up assessment between three and five months after hospital discharge. Patients with cardiac arrhythmias (including frequent atrial and ventricular premature contractions, atrial flutter, atrial fibrillation, supraventricular tachycardia and atrioventricular block (second degree or higher)), pacemaker, bradycardia (heart rate < 50 bpm) or tachycardia (heart rate > 110 bpm) were excluded, as these conditions may influence reliable HRV measurements. In case no ECG was obtained during admission, patients were also excluded.

### 2.2. Data Collection

Data were retrospectively collected from patients’ medical records based on the WHO-COVID case record form [23] and stored in Castor Electronic Data Capture (EDC), an ISO-certified protected database. This included information on patient characteristics such as sex, age, body mass index (BMI) and comorbidities (e.g., chronic heart disease including all heart diseases except hypertension, diabetes, chronic lung disease) and clinical data such as oxygen therapy, date of onset and length of stay. All patients were invited for a standardized outpatient follow-up assessment three months after discharge. During this visit, pulmonary function tests (PFTs), ECG and chest CT (computed tomography) scan were performed, followed by an appointment with an internal medicine specialist and a pulmonologist.

PFTs were carried out according to the guidelines of the European Respiratory Society on a MasterScreenTM Body and MasterScreenTM PFT (PanGas, Dagmersellen) using the SentrySuite V3.0.5 software. The following tests were performed: forced expiratory volume in one second (FEV1), forced vital capacity (FVC), total lung capacity (TLC), diffusion capacity of the lung for carbon monoxide (DLCO), Tiffeneau index (FEV1/FVC), maximal respiratory expiratory and inspiratory pressure (PE and PI). All PFT parameters were expressed as percentage of predicted normal values. Impaired lung function was defined as FEV1/FVC < 70% and <80% predicted for all other parameters [24]. Furthermore, during the outpatient assessment, patients were questioned whether they still experienced symptoms related to COVID-19. No validated questionnaires were used, and therefore, only the five most frequently reported symptoms (e.g., fatigue, exertional dyspnea, cough, chest pain and palpitations) were used in this study. Patients were then divided into four categories based on the number of reported symptoms: none, one, two, or three to five.

### 2.3. Heart Rate Variability

HRV was analyzed from one 12-lead 10 s ECG (150 Hz) obtained closest to the day of admission. In case multiple ECGs were taken upon admission, the ECG with the best quality was used for analysis. ECGs were assessed for eligibility based on the predefined exclusion criteria. In case of doubt, a second reviewer (M. T. A. Strous) screened the ECGs.

The time between two adjacent heartbeats, the R-R interval, was measured in lead II with an accuracy of 0.2 ms, using the MUSETM-ECG system (General Electric Company, Boston, MA, USA). The HRV was expressed by the time-domain HRV parameters including the standard deviation of NN intervals (SDNN), reflecting total HRV, and the root mean square of successive differences (RMSSD), mostly reflecting vagal activity [25]. These were calculated using common formulas [26]:(1)$$\mathrm{SDNN}=\sqrt{\frac{1}{n-1}\;\sum_{i=1}^{n}{\left(RR_{i}-RR_{mean}\right)}^{2}}$$
(2)$$\mathrm{RMSSD}=\sqrt{\frac{1}{n-1}\;\sum_{i=1}^{n-1}{\left(RR_{i+1}-RR_{i}\right)}^{2}}$$

Power spectral analysis requires longer ECG recordings and was therefore not feasible in this study [27]. The cut-off point for dividing low and high HRV groups was 8 ms, based on a median SDNN of 7.0 (IQR 4.6–10.6) and a median RMSDD of 7.7 (IQR 4.7–11.3), which was similar to a study that predicted mortality and ICU referral in hospitalized COVID-19 patients [28]. Although HRV measurements may be variable (and sometimes situational) within individuals, chronically low HRV measurements are generally not favorable for several conditions, including COVID-19 [28]. Furthermore, when assessing HRV from a 10 s ECG, RMSDD is considered the most reliable parameter [27,29].

### 2.4. Statistical Analysis

Descriptive statistics were used to provide an overview of confounding and main study variables (including age, sex, comorbidities, oxygen therapy during hospitalization, duration of hospital stay, time between discharge and assessments, time between onset of symptoms and ECG, HRV and outcomes). Non-normally distributed variables were presented as medians with interquartile range (IQR) and compared between HRV groups using the Mann–Whitney U test. All other continuous variables were presented as mean and standard deviation and compared between HRV groups using the unpaired *t*-test. All categorical variables were compared between HRV groups using chi-square statistics and described by the absolute number and corresponding percentage.

The associations between the number of symptoms with different PFT outcomes and with HRV were analyzed by multivariable logistic regression and multinomial logistic regression models, respectively. In addition to age and gender, other patient or treatment-related factors were considered as possible confounders if they showed relevant differences (*p* < 0.1) between HRV categories after performing a Mann–Whitney U test, *t*-test or chi-square test. Patients were divided into age groups because pulmonary function and HRV both decrease with rising age [25,30]. The female gender was included as a confounding variable because of the correlation with reduced pulmonary function [31,32], higher risk of developing PCC [9] and lower mean SDNN index values [33]. All data were analyzed by using IBM SPSS statistics, version 24.0. Results were considered significant with a two-tailed *p*-value ≤ 0.05.

### 2.5. Ethical Considerations

A waiver for medical ethical review was provided by the Medical Ethical Committee of Maastricht University Medical Center (approval number: METC 2021-3059).

## 3. Results

### 3.1. Study Population

Between February and December 2020, 577 patients were hospitalized for COVID-19. In total, 268 patients were lost to follow-up because of death during hospitalization or before the planned assessment (*n* = 159), readmission (*n* = 4), cancellation of the outpatient clinic visit (*n* = 12) or follow-up by telephone (*n* = 20), transfer to rehabilitation units (*n* = 8), or patients were transferred from other hospitals because of limited capacity and were followed up with by their local hospital (*n* = 65) (Figure 1). Of the remaining 309 patients, 138 patients were excluded due to ICU admission (*n* = 61), incomplete PFTs (*n* = 5) or because no reliable HRV measurement could be performed (*n* = 72). HRV analysis was not possible because of poor quality of ECG (*n* = 4), pacemaker (*n* = 7), no ECG was available (*n* = 3), tachycardia (*n* = 13) or arrhythmia (*n* = 45). Eventually, 171 patients were included for analysis.

### 3.2. Baseline Characteristics

The mean age at diagnosis of the participants was 67 years (SD ± 11) and 61% were male (Table 1). The most common comorbidity was hypertension (42%), followed by obesity (35%), type I and II diabetes (25%), chronic heart disease (23%) and chronic obstructive pulmonary disease (22%). The median length of hospital stay was six days (IQR 3–9 days), during which 149 patients (87%) received oxygen therapy.

All ECGs were obtained within ten days after admission, and 91% were obtained on the day of admission. In two cases, ECGs from a previous emergency visit with COVID-19 (three and eleven days before admission) were used. The median time from discharge to the outpatient follow-up assessment was 119 days (IQR 101–141). PFTs were performed after a median of 113 days (IQR 98–134) following discharge. The median period between onset of symptoms to obtaining the ECG was eight days (IQR 7–11 days).

Just over half of the patients had HRV below ≤8 ms (SDNN ≤ 8: 58% and RMSSD ≤ 8: 53%). In the low HRV group, patients were younger (65 vs. 69 years, *p* < 0.031) and a higher proportion of patients had diabetes (RMSDD: 31% vs. 17%, *p* < 0.036) and a higher proportion received oxygen therapy during hospitalization (SDNN: 91% vs. 81%, *p* < 0.028). Therefore, these variables were added as confounders in the multivariate analyses.

### 3.3. Pulmonary Function Tests

The most commonly impaired PFT was decreased DLCO (41%), followed by reduced PE (37%), PI (31%), FEV1 (23%), FVC (15%), TLC (12%) and FEV1/FVC (8%). Multivariable logistic regression (Table 2) revealed that patients with high HRV (indexed as RMSDD > 8) measured upon admission had a significantly lower probability of reduced TLC (O.R. = 0.28, 95% CI: 0.09–0.90, *p* < 0.033). Due to limited statistical power and subsequently wide 95% confidence intervals, other potential confounders such as pre-existent lung disease or diabetes were not considered in the analyses.

### 3.4. Number of Symptoms

After a median of 119 days following discharge, 139 out of the 171 patients (81%) reported at least one symptom. The most frequently reported symptoms were fatigue (64%) and exertional dyspnea (56%), followed by cough (32%), chest pain (22%) and palpitations (10%). Proportions of these reported symptoms were equally divided between HRV groups, as shown in Figure 2.

Of the total 171 patients, 12 had to be excluded for the analysis on the number of reported symptoms, because symptom experience was not reported for all predefined symptoms. Of the remaining 159 patients, 23% experienced only one symptom, 25% experienced two symptoms and 32% experienced three to five symptoms (Table 3). Multinomial logistic regression revealed that HRV measured at admission was not associated with the number of reported symptoms between three and five months after COVID-19 hospitalization. Again, the confounding variables of diabetes and oxygen therapy during hospitalization were not added to analysis due to the broad confidence intervals.

## 4. Discussion

Persistence of symptoms beyond three months after an infection with COVID-19 is often reported and is not necessarily associated with the severity of infection. In this study, pulmonary function impairment and persistent symptoms were present in the majority of patients between three to five months after discharge for COVID-19 hospitalization. Although several studies previously suggested that persistent autonomic dysfunction may be (partly) responsible for the long-term consequences of COVID-19, we did not find an association between HRV measured upon admission and persistent pulmonary function impairment or the number of reported symptoms post-COVID-19 hospitalization.

Previous research revealed that patients with PCC most frequently suffer from decreased DLCO [4,34,35], followed by reduced FEV1 and FVC [34,36], and TLC [4,34,37]. Studies in which PFTs were obtained approximately three months after COVID-19 infection show similar pulmonary function impairments as our study, with abnormalities in DLCO for 16–57%, in FVC for 5–24%, in TLC for 10–27% and in FEV1 for 6–30% [17,34,35,36,37]. The wide variation in pulmonary function impairment rates could likely be explained by differences in the study population, as some studies only included non-hospitalized patients or severe cases. In our study, lung function was presented as % predicted values. Only one study presented normality for pulmonary function parameters based on the lower limit of normal (LLN) [38]. LLN equals the 5th percentile of a healthy, non-smoking population and has a higher validity than the 80% predicted value, especially in older patients [24,39]. This possibly explains why the LNN study found a lower prevalence of pulmonary function impairments.

We found no association between HRV measured on admission and pulmonary function impairment between three and five months after discharge for COVID-19 hospitalization. Post hoc power analyses showed that our sample size was not sufficient to show ORs closer to 1.00 than 0.63. Therefore, the association may still exist, but may be weaker than can be demonstrated with this study size. If present, its clinical relevance could be questioned.

At least two other studies investigated the correlation between HRV and pulmonary function impairment after the acute phase of COVID-19. The first study included 18 patients and did not find a correlation between HRV measured during follow-up and FVC [40]. The second study measured HRV (indexed as SDNN) six months post-discharge and found an association with diffusion dysfunction, but not with ventilation dysfunction (defined as FEV1/FVC < 70%) [41]. It could be argued that HRV only has a predictive value within in a shorter period of time (weeks rather than months). However, it was previously found that HRV measured between three and six months after an acute coronary syndrome still had a prognostic value within nine years following hospitalization [42].

The rate of persistent symptoms reported in our study cohort is in line with the study performed by Evans et al., who distributed a symptom questionnaire to 861 patients at a median of six months (IQR 5–7 months) post-discharge [8]. In our study, multinomial logistic regression did not show a correlation between HRV measured upon admission and the number of reported symptoms at three to five months after discharge. Furthermore, ORs differed between RMSDD and SDNN groups. Two previous studies determined that only RMSDD is a reliable parameter for assessing HRV from a 10 s ECG [27,29], which may explain the differences in ORs. Furthermore, in multiple cohorts, it was observed that compared to healthy, uninfected individuals, HRV was lower in patients 30 days after COVID-19 infection to six months post-discharge [40,43,44]. Moreover, COVID-19 patients with orthostatic hypotension had a significantly lower HRV (indexed as RMSDD) compared to those without [43]. These findings suggest that HRV can be considered as a marker for cardiovascular dysautonomia. In contrast, however, another study revealed that patients who experienced symptoms for >3 and >6 months had higher HRV than those who experienced symptoms for ≤3 months [44].

This study was subject to a number of limitations. The variation in time from discharge to assessment was broad and no validated questionnaires were used to assess symptom burden. This might have influenced the prevalence of the number of reported symptoms. However, the rate of persistent symptoms found in this study is in line with previous work. Moreover, there were a lack of baseline values for pulmonary function and symptomology prior to COVID-19 hospitalization. Due to the retrospective design, we did not have full control over measurement of confounders. Nevertheless, the database included extensive information and suspected major confounding variables were added to analysis. This study only included hospitalized patients, which may have led to selection bias as patients of older age are more likely to be admitted than younger patients. Because low HRV is usually associated with frailty in the elderly, it is remarkable that in this overall 60+ population, those in the low HRV group are on average five years younger than those with a high HRV. This is further substantiated by the fact that the low HRV group was more likely to have received oxygen therapy and have a history of diabetes. In summary, these findings suggest that HRV is a possible marker for in-hospital frailty of patients, although the relatively young patients in the low HRV group and those in acute need of oxygen therapy were not necessarily the patients who experienced the highest rates of lung function impairments post-discharge.

There were also a number of strengths to this study. To the best of our knowledge, this was one of the first studies to investigate the association of HRV with pulmonary function and the number of reported symptoms in the post-COVID-19 period. Because HRV measurement is easy and non-invasive, it could be easily implemented in daily clinical practice. If proven useful, this may aid clinicians in identifying frail patients at risk of prolonged recovery. It may also have therapeutic consequences, as results from clinical studies using vagus nerve stimulation in illnesses with excessive inflammation, such as Crohn’s disease and rheumatoid arthritis, are encouraging [45]. However, these preliminary findings warrant further research and do not yet support widespread implementation in clinical practice. It may also only be of added value in specific patient groups.

Future studies should further explore the predictive value of HRV in PCC by using larger sample sizes, baseline measurements of pulmonary function expressed with LLN and validated questionnaires to assess clinical symptomatology. Because chronic heart disease and COPD may potentially affect HRV and lung function test results, it may be valuable to perform HRV sub-analyses in these groups in studies with larger sample sizes. Furthermore, we suggest future studies to routinely measure HRV in order to assess if increasing HRV during the post-COVID-19 period is associated with faster recovery.

## 5. Conclusions

In conclusion, a single HRV measurement upon admission was not associated with pulmonary function impairment or persistent symptoms three to five months after hospitalization for COVID-19.

## Figures and Tables

**Figure 1 sensors-23-02473-f001:**
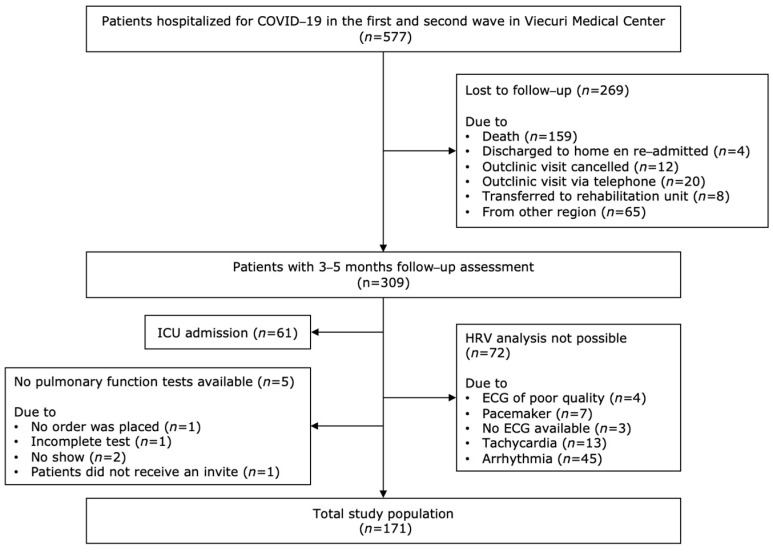
Study flow chart.

**Figure 2 sensors-23-02473-f002:**
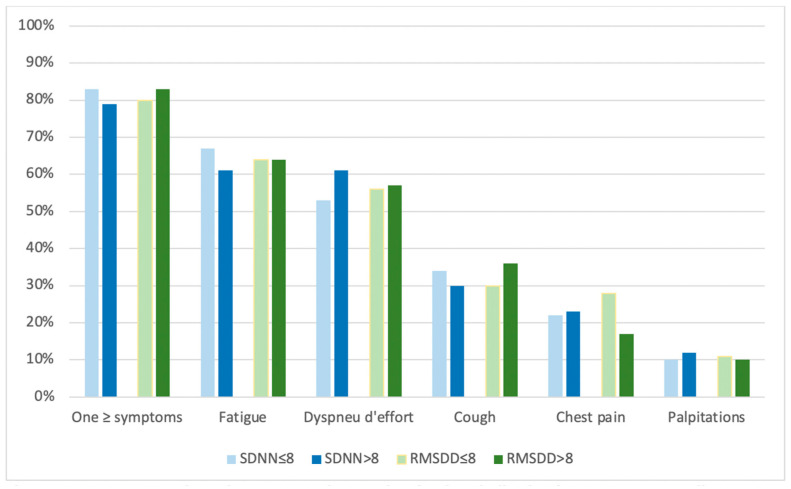
Symptom experience between 3 and 5 months after hospitalization for COVID-19 according to heart rate variability. Note: SDNN = standard deviation of normal to normal heartbeat intervals; RMSDD = root mean square of successive differences between adjacent heartbeats. Both SDNN and RMSSD cut-offs are in msec.

**Table 1 sensors-23-02473-t001:** Descriptive data of included patients according to heart rate variability.

Variables	Total	SDNN ≤ 8	SDNN > 8	*p*-Value	RMSSD ≤ 8	RMSDD > 8	*p*-Value
N	171	99 (58)	72 (42)		90 (53)	81 (47)	
Age in years, mean ± SD	67 ± 11	65 ± 11	69 ± 11	0.031	65 ± 11	69 ± 11	0.031
Sex, *n* (%)				0.802			0.389
Male	105 (61)	60 (61)	45 (63)		58 (64)	47 (58)	
Female	66 (39)	39 (39)	27 (38)		32 (36)	34 (42)	
Comorbidities, *n* (%)							
Chronic obstructive pulmonary disease	37 (22) [7]	18 (19)	19 (28)	0.194	17 (19)	20 (26)	0.285
Chronic heart disease	37 (23) [8]	23 (24)	14 (21)	0.586	16 (18)	21 (28)	0.160
Hypertension	72 (42)	41 (41)	31 (43)	0.830	36 (40)	36 (44)	0.557
Obesity (BMI ≥ 30)	59 (35) [2]	35 (36)	24 (33)	0.711	29 (33)	30 (37)	0.578
Diabetes	42 (25) [2]	26 (26)	16 (22)	0.545	28 (31)	14 (17)	0.036
Hospital stay							
Oxygen therapy, *n* (%)	149 (87)	91 (91)	58 (81)	0.028	80 (89)	69 (85)	0.470
Days from admission to discharge	6 {3–9}	6 {3–10}	6 {3–9}	0.528	6 {3–10}	6 {3–9}	0.507
Days from discharge to follow-up assessment	119 {101–141}	120 {101–144}	118 {102–141}	0.857	121 {102–144}	117 {101–141}	0.909
Days from discharge to lung function assessment	113 {98–134}	113 {97–137}	113 {98–134}	0.939	112 {97–138}	113 {98–133}	0.949

Normal distributed data was compared by using unpaired t-test and presented as mean ± SD. Non normal-distributed data was analyzed by using Mann-Whitney U test and presented as median {IQR}. Categorical variables were compared between HRV-groups by using Chi-squared tests and described by the absolute number, (corresponding percentage) and [missing values]. Note: SDNN = standard deviation of normal to normal heartbeat intervals; RMSDD = root mean square of successive differences between adjacent heartbeats. Both SDNN and RMSSD cut-offs are in msec.

**Table 2 sensors-23-02473-t002:** Odds ratios with confidence intervals for abnormal pulmonary function between 3 and 5 months after hospitalization with COVID-19 using multivariate logistic regression.

		FEV1	FEV1/FVC *	FVC	TLC	DLCO	PI	PE
<80% pred	*n*, (%)	40 (23)	14 (8) (1)	26 (15)	19 (12) [6]	66 (41) [8]	50 (31) (1)	60 (37) (1)
SDNN > 8	Unadjusted	1.18 (0.57–2.38)	1.02 (0.34–3.10)	0.69 (0.29–1.65)	0.31 (0.10–0.99)	1.24 (0.66–2.33)	0.93 (0.47–1.83)	1.01 (0.53–1.92)
	Model 1	1.07 (0.51–2.22)	0.94 (0.31–2.90)	0.69 (0.29–1.65)	0.32 (0.10–1.02)	1.08 (0.56–2.11)	0.82 (0.40–1.66)	0.89 (0.44–1.81)
	*p*	0.865	0.920	0.400	0.055	0.813	0.580	0.755
RMSDD > 8	Unadjusted	1.15 (0.57–2.33)	1.56 (0.52–4.70)	0.94 (0.41–2.18)	0.27 (0.08–0.84)	0.91 (0.51–1.80)	0.98 (0.50–1.92)	1.02 (0.54–1.94)
	Model 1	1.03 (0.50–2.15)	1.41 (0.46–4.31)	0.94 (0.41–2.19)	0.28 (0.09–0.90)	0.88 (0.46–1.70)	0.92 (0.46–1.85)	1.00 (0.50–2.00)
	*p*	0.933	0.552	0.891	0.033	0.705	0.811	0.994

Categorical variables were presented as absolute number, (corresponding percentage) and [missing value]. Model 1 was adjusted for age groups and gender. * A threshold of 80% predicted was used as a cut-off point for all pulmonary function tests, expect FEV1/FVC, for which a cut-off point of 70% was used. Note: SDNN = standard deviation of normal to normal heartbeat intervals; RMSDD = root mean square of successive differences between adjacent heartbeats. Both SDNN and RMSSD cut-offs are in msec.

**Table 3 sensors-23-02473-t003:** Odds ratios with confidence intervals for symptom experience between 3 and 5 months after hospitalization with COVID-19 using multinomial logistic regression.

	Number of Symptoms	None	One	Two	Three to Five
		32 (20)	36 (23)	40 (25)	51 (32)
SDNN > 8	Unadjusted	Reference category	2.29 (0.84–6.28)	0.72 (0.28–1.84)	1.26 (0.52–3.07)
	Model 1		2.37 (0.86–6.53)	0.72 (0.28–1.84)	1.23 (0.50–3.00)
	*p*		0.095	0.493	0.657
RMSDD > 8	Unadjusted		1.09 (0.42–2.85)	1.03 (0.42–2.50)	1.03 (0.42–2.50)
	Model 1		1.10 (0.42–2.91)	0.99 (0.40–2.43)	0.99 (0.40–2.43)
	*p*		0.848	0.978	0.978

Model 1: Adjusted for age groups and gender. Note: SDNN = standard deviation of normal to normal heartbeat intervals; RMSDD = root mean square of successive differences between adjacent heartbeats. Both SDNN and RMSSD cut-offs are in msec.

## Data Availability

The datasets used and/or analyzed during the current study are available from the corresponding author on reasonable request.

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
