# Peer review of "Association of Heart Rate Variability with Pulmonary Function Impairment and Symptomatology Post-COVID-19 Hospitalization"

_sensors, 2023, doi:10.3390/s23052473_

Round 1
Reviewer 1 Report
The work is interesting and up-to-date.
The abstract should perhaps be written as in templates with Background, METHODS, RESULTS, and CONCLUSIONS.
The discussions are too long and present a series of aspects that should be discussed in the Results section. (eg line 229-231; line 251-259; line 270-271; 280-300)
The Conclusions of the study must also be separated.
The tables and images must be discussed in the results, each separately, not a subchapter containing tables and figures.
A happy new year with accomplishments!!!!
Reviewer 2 Report
Please provide the reference(s) where HRV is stated/posited as a way to measure vagal nerve activity.
Reviewer 3 Report
Please find attached file.

Round 2
Reviewer 2 Report
Authors have made sufficient improvement of the manuscript in all fields were corrections were requested.